# GPT Carry-On: Language Model Customization Made Scalable by Growing-In-Depth Training

## Abstract

Modern large language foundation models (LLM) have now entered the daily lives of millions of users. We ask a natural question whether it is possible to customize LLM for every user or every task. From system and industrial economy consideration, general continue-training or fine-tuning still require substantial computation and memory of training GPU nodes, whereas most inference nodes under deployment, possibly with lower-end GPUs, are configured to make forward pass fastest possible. We propose a framework to take full advantages of existing LLMs and systems of online service. We train an additional branch of transformer blocks on the final-layer embedding of pretrained LLMs, which is the base, then a carry-on module merge the base models to compose a customized LLM. We can mix multiple layers, or multiple LLMs specialized in different domains such as chat, coding, math, to form a new mixture of LLM that best fit a new task. As the base model don't need to update parameters, we are able to outsource most computation of the training job on inference nodes, and only train a lightweight carry-on on training nodes, where we consume less than 1GB GPU memory to train a 100M carry-on layer on 30B LLM. We tested Qwen and DeepSeek opensourced models for continue-pretraining and got faster loss convergence. We use it to improve solving math questions with extremely small computation and model size, with 1000 data samples of chain-of-thoughts, and as small as 1 MB parameters of two layer layer carry-on, and the results are promising.

## 1 Introduction

The rapid development of large language models (LLMs) like GPT-4 Achiam et al. (2023) and DeepSeek-R1Guo et al. (2025) has grown from a foundational model of natural language processing (NLP) downstream tasks, to achieve promising AI capabilities in numerous fields. They are pretrained on massive datasets, then generalize across tasks through in-context learning or lightweight prompting Brown et al. (2020). However, whether several foundation models are enough for all users, is at question. After all, millions of users have their own professions, specialties, tasks, such as medical diagnostics, legal analysis, and they should have language preferences of themselves. We imagine if there is a method to customize LLMs for each individual or tasks like personalized news feeding. This idea requires further training and diverging to numerous new versions, and deploying them to service is definitely difficult.

On the hardware and system side, this introduces too much cost for LLM providers to train so many versions. The provider generally use *inference devices* for online services. But training jobs need to be placed on *training devices* due to the much higher computational and memory demands. To be specific, training involves both forward and backward passes. Gradients are computed during the backward pass and are essential for updating the model weights. Optimizers (e.g., Adam, SGD) are used to update model weights based on the computed gradients. Inference only involves the forward pass, and doesn't need to store gradients and optimizer states. During training, high precision (e.g., FP32 or mixed precision with FP16) is typically required to ensure stable gradient calculations and accurate weight updates. During inference, quantization Achiam et al. (2023); Lin et al. (2024) like INT8 is often applied to reduce the model size and speed up computation, and software stack is optimized to batch user request and reuse key-value cache Kwon et al.

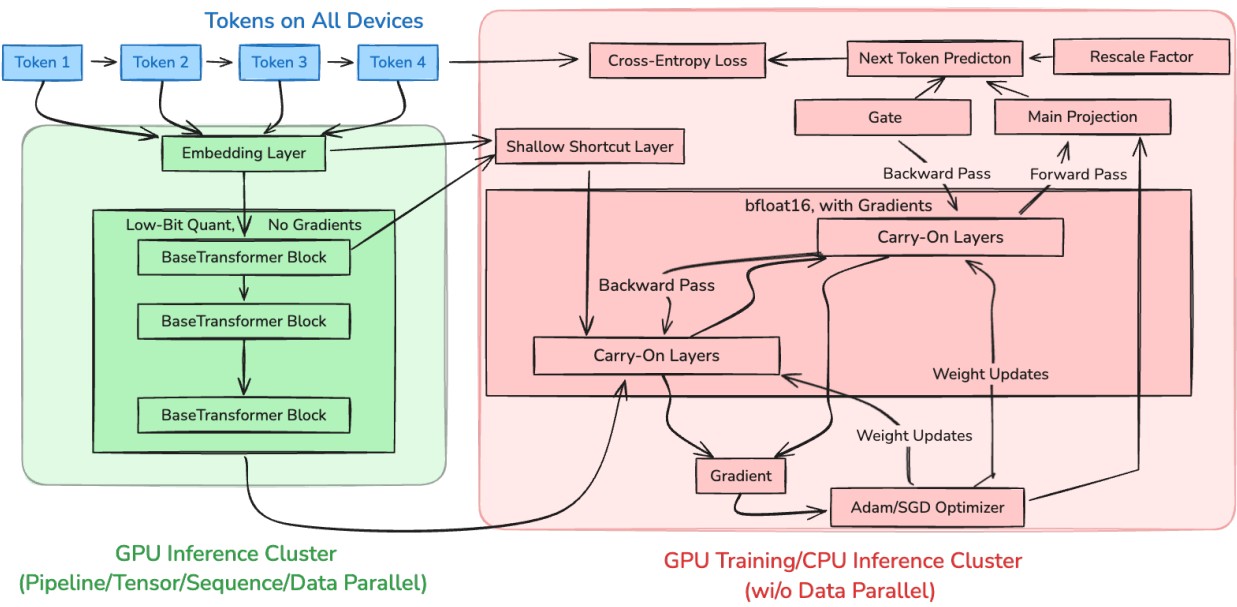

Figure 1: The architectural design of the transformer-carry-on

(2023). This encourages the use of GPUs with smaller memory capacities or specialized inference accelerators, and reserve most of the advanced GPUs to training nodes, which makes sense from commercial perspective.

On the algorithm side, continue-training of LLM are going to infuse personal or task related corpus, but unfortunately this doesn't always improve the intelligence level nor the capability on this specific task, since these new corpus may not meet the quality standard of delicate pretraining and supervised fine-tuning datasets from original LLM provider like OpenAI. Full-parameter training, or even parameter-efficient adaptor layersHoulsby et al. (2019) like Low-Rank Adaptation (LoRA) Hu et al. (2022), are designed to change the entire picture of the LLM output, Raffel et al. (2020), could possibly erode the model's general-purpose knowledge learned through millions or billions of GPU hours, known as forgetting. These backfires inspire us to compose a methodology that finds good balance between customization and maintaining state-of-the-art general intelligence, and to leave the decision to users.

Our framework rethinks adaptation as building more transformer on top of the pretrained LLM transformers, which we will refer to as the base LLM. The training process can be taken in two different GPUs: to run a forward pass of the base model in the inference GPU node and corresponding specification, which targets at extreme acceleration, utilizing low-bit quantization Aminabadi et al. (2022); Sheng et al. (2023); Zhao et al. (2024), tensor-parallelism, or even specialized hardware Pope et al. (2023). The inference node then compresses the final-layer embeddings to transmit to another training node, which may only have lower-end GPU with smaller memory, and this node train a lightweight transformer carry-on to finish the task of next-token prediction. This adaptor will also compensate for the precision loss of the inference acceleration.

## 2 Background

Transformer-based large language models (LLMs) typically represent the input sequence of tokens as a set of token embeddings. Suppose we have a sequence of tokens $S_1, S_2, \cdots, S_n$. Each token $S_n$ is mapped to a $d$-dimensional embedding vector $\mathbf{x}_i \in \mathbb{R}^d$. The entire sequence of embeddings can be represented as a matrix $\mathbf{X} = [\mathbf{x}_1, \mathbf{x}_2, \cdots, \mathbf{x}_n]^T \in \mathbb{R}^{n \times d}$, where $n$ is the sequence length and $d$ is the embedding dimension. The key of the Transformer is the attention mechanism. For a given input sequence $\mathbf{X}$, we first compute three matrices: the query matrix $\mathbf{Q}$, the key matrix $\mathbf{K}$, and the value matrix $\mathbf{V}$. The self-attention scores between tokens are

computed as

$$A = \text{softmax}(\frac{\mathbf{Q}\mathbf{K}^T}{\sqrt{d_k}}) \in \mathbb{R}^{n \times n} \tag{1}$$

When using multiple attention heads ($h$ heads), we concatenate the outputs of each head. Layers of attentions, normalizations, and multilayer perceptrons (MLP) and residual connections stack to make a large transformer model. In our paper, we will use $h$ to represent the pretrained transformer model, and with $w$ as its parameters. We use $\mathbf{x}^L = h_w(S_{[1:n]})$ represents the embedding of a single token within a sequence at the highest layer. These embeddings capture information from all tokens before it and the token itself. The final projection, i.e. the highest level linear layer, takes the embedding $\mathbf{x}^L$ to the output embedding, which predicts the next token in the sequence,

$$\mathbf{y}_{n+1} = W_{\text{output}}\mathbf{x}_n^L, \tag{2}$$

where $W_{\text{output}}$ is the weight matrix of the original linear projection in the decoder.

Training state-of-the-art LLMs involves significant computational and memory requirements. In mixed precision training (FP16 or BF16), each optimized parameter requires 2 bytes for the weight, 4 bytes for the gradient (Float32), and 8 bytes for the optimizer states (e.g., Adam optimizer, which maintains two states of Float32). This results in approximately 14 bytes per parameter and it may exceed GPU memory capacity since the LLM model size was designed to maximize memory usage and hit the roof. Additionally, intermediate activations during the forward and backward passes can add up. Inference however, has significantly lower memory and computational requirements compared to training, as it only involves the forward pass and does not require storing gradients or optimizer states. In FP16 precision, the 7B LLM model weights occupy approximately 14GB (7B × 2 bytes), which can fit comfortably within the memory of a single NVIDIA A10G (24GB) GPU. To further optimize memory usage and increase throughput, quantization techniques such as INT4 can be applied, reducing the model size to around 7GB (7B × 1 byte). This allows for dynamic batching, where multiple inference requests are grouped together to maximize GPU utilization. Optimized inference frameworks like TensorRT-LLM or vLLM Kwon et al. (2023) can be used to achieve high throughput.

## 3 Carry-On-Training Implementation

Our main goal is to reuse the inference node in industrial application when inference GPU nodes are specified for extreme acceleration and therefore sacrificing precision, and these nodes mostly will serve base version LLM. To bridge the inaccurate embedding space and the final target, we introduce a carry-on-training framework for LLM. Unlike LoRA Hu et al. (2022), which injects low-rank matrices into the base model weights, so it is largely affected by its architecture, the carry-on can specify its own form and parameters, number of layers, layer dimensions. The carry-on might be a simple linear transformation, or a medium scale mixture-of-experts (MoE) transformer. LoRA backpropagates gradient to the base model, while carry-on adaptor doesn't, so it can be trained independently, as long as it access the highest layer embedding.

Aside from LLM, the training philosophy has been different: building more layers or functions on top of existing representations, with or without training existing layers. Classical machine learning like boosting algorithms Freund & Schapire (1997) in combine weak learners (e.g., decision trees) through weighted linear combinations, and ResNet inherits essence from prior works and achieved greatness through simplicity. For generative models, *Stable Diffusion* Rombach et al. (2022) train a variational autoencoder (VAE) to compress images into low-dimensional latents in an independent stage, followed by a transformer trained on these latents. This separation reduces computational complexity while maintaining fidelity. Speech recognition (ASR) system like *Whisper* Radford et al. (2023) builds a higher-level transformer to adapt the system to specialized tasks or domains, based on acoustic transformer, CNN network, and mel spectrum. These paradigms share a common theme: building atop existing representations rather than modifying them directly. Translating to LLMs, we hypothesize that the pretrained LLMs at this date has well passed most internet user's ability to further improve in general, our invasive modification to each layer could be non-necessary.

Following the thoughts, we propose the *GPT Carry-On Trainer* demonstrated in Fig 1. It is implemented on two different types of GPU nodes, training and inference nodes, where the inference nodes are the ones serving LLM online users, without any changes to its software configuration as well. The training starts from

the inference nodes loading and running forward pass of the base LLM which we want to train from. There is a bridge function afterwards to compress the embedding of the base model $\mathbf{x}^L \in \mathbb{R}^D$. It is transformed into representation of even smaller bits $\sigma(x)$, through a deterministic like lower-bits quantization, or learned operator like vector quantization or linear projection to lower dimensions. The embedding along with original tokens are transfered to training nodes.

- Inference Nodes Forward Pass:    $x \leftarrow h_w(S)$

- Bridge Function:    $\sigma(\mathbf{x}) = (W_{\text{align}}\mathbf{x}),$

- Communication:    $\sigma(x), S \rightarrow \text{training-nodes},$

- Training Nodes Optimize:    $\min \mathcal{L}_{\text{pred}}(x, S)$

Then we train more neural network layers on top of the embedding, and they may or may not be transformer blocks, ResNets, or even RNN layers, to do the next token prediction. We use $f$ to represent the composite neural network, and use $\theta$ to represent its parameters, the prediction loss $\mathcal{L}_{\text{pred}}$ depends on the next token $\mathbf{S}_{\text{next}}$ using a softmax layer. The architecture of transformer-carry-on is designed to explore the correct balance between the base LLM's knowledge and the customization task, therefore we use a rescale factor to control this balance. To add delicate control, we also use finer-level gate to control each element of the embedding, of their contribution to final output.

$$\Delta\mathbf{x} = \underbrace{g_\theta(\sigma(x), S)}_{\text{Gate}} \odot \underbrace{f_\theta(\sigma(x), S)}_{\text{Main Func}} \tag{3}$$

$$\mathbf{y}(\alpha) = W_{\text{pred}}(\alpha\Delta\mathbf{x} + \mathbf{x}) \tag{4}$$

where $W_{\text{pred}} \in \mathbb{R}^{V \times D}$ is the projection to vocabulary space, and it could be a new learnable parameter or inherited from the base foundation model, and $V$ is the vocabulary size.

The parameter $\alpha$ is the rescale factor, a non-negative value, control the intensity of carry-on to the final logits; and $g$ generate a group of gate values between 0 and 1, controls the contribution of each element. The gate and the carry-on branch can share most layers, and only differs in the activation function their own final linear layer, where the gate function uses a sigmoid activation. To train the carry-on layers, as the base LLM embedding predicts well, the carry-on layers won't get strong derivative signals as first, we can set $\alpha$ a big value like 5.0 from start, and gradually decrease it, and will determine its final value by validation.

As we try to not train base LLM that changes its parameters, we are strongly motivated to harvest information from all possible sources to boost or tune the output of the highest layer, e.g. ensembling different LLMs by mixing their highest level embedding. Another approach is to take more advantages of information flow from shallow to deep layers of deep neural networks. We can see the contribution of these shallow layer, although discarded, from information theory Cover (1999): the data processing inequality states that if a random variable $X$ is transformed into $Y$ and then $Y$ is transformed into $Z$, then

$$I(X; Z) \leq I(X; Y), \quad X \rightarrow Y \rightarrow Z$$

where $I$ denotes mutual information. Based on the inequality, consider the sequence of token embeddings in a transformer, where $\mathbf{x}^0$ represents the embedding at the shallow layer and $\mathbf{x}^L$ represents the embedding at the highest layer. Each layer is the condition of key-value (KV) pairs of deeper layers as Eq.(1). These KV pairs are used in subsequent layers for computing attention over the entire sequence, enabling the model to interchange information between tokens. The highest layer $\mathbf{x}^L$ is primarily optimized to predict the next token, causing it to discard information that is not directly relevant to this immediate prediction. In contrast, shallow layer $\mathbf{x}^0$ contains richer information about past tokens, before processing and passing them to the deep layer $\mathbf{x}^L$. Let $S_{next}$ be the next tokens in the sequence $S$, we have

$$I(\mathbf{x}^0; S_{next}) < I(\mathbf{x}^L; S_{next}), I(\mathbf{x}^0; S_{past}) > I(\mathbf{x}^L; S_{past})$$

by analysis above. Then we also have $I(\mathbf{x}^0; S) > I(\mathbf{x}^L; S)$ as shallow layers provides KV pairs to deeper layers to generate more future tokens. From this perspective, we see the shallow layers provide complementary or orthogonal information to deeper layers, so the carry-on layers benefit from more information sources.

Insipred by this, instead of building the carry-on transformer mainly based on highest layer embedding, we could also use shallow shortcut layers from original LLM, and fuse embedding from different layers, e.g. we take 32-th layer and 0-th layer of 7B LLM and take element-wise average, as the dimension of embedding won't change after this linear operation.

```python
# Load 4-bit base model with tensor parallelism
base_model = AutoModelForCausalLM.from_pretrained(
    model_path,
    load_in_4bit=True,
    torch_dtype=torch.float16
).to(inference_device)
base_model.eval()

for param in base_model.parameters():
    param.requires_grad = False

# Trainable carry-on layers
class CarryOnLayers(torch.nn.Module):
    def __init__(self, params):
        super().__init__()
        self.bridge = torch.nn.Linear()
        self.layers = TransformerLayers()
        self.gate = torch.nn.Linear()
        self.proj = torch.nn.Linear()
        self.classifier = torch.nn.Linear()
        self.alpha = 0.5

    def forward(self, x_deep, x_shallow):
        x = self.bridge(x_shallow) + x_deep
        for layer in self.layers:
            x = layer(x)
        gated = Sigmoid(self.gate(x))
        x = gated * self.proj(x)
        x = x_deep + self.alpha * x
        y = self.classifier(x)
        return y

carry_on = CarryOnLayers().to(train_device)
optimizer = optim.Adam(carry_on.params)
scaler = GradScaler()

# Training loop
def train_step(input_ids, targets):
    # Base model inference
    with torch.no_grad():
        outputs = base_model(
            input_ids.to(inference_device),
            output_hidden_states=True
        )
    deep = quantize(outputs.layer[-1])
    shallow = quantize(outputs.layer[0])
    deep = deep.to(train_device)
    shallow = shallow.to(train_device)
    # Carry-on training
    optimizer.zero_grad()
    with autocast(dtype=torch.bfloat16):
        logits = carry_on(deep_layer, shallow_layer)
        loss = cross_entropy(
            logits[:, :-1], targets[:, 1:]
        )
    scaler.scale(loss).backward()
    scaler.step(optimizer)
    scaler.update()
```

```
59        return loss.item()
```
Listing 1: A Simple Pseudo Implementation

In this way, we are able to outsource most of the training computation and memory usage to inference nodes, and the training nodes only need to receive these embedding vectors, and run forward/backward and optimizer for a significantly smaller transformer. After the carry-on layers are trained, we can move these layers to inference nodes so all layers can finish within one node, by consuming a considerably small GPU memory.

## 4  Customization v.s. Generalization

We will further determine the optimal $\alpha$ through the evaluation process. The evaluation of the LLM could be based on diverse kind of criterion. The criterion could be the same cross-entropy loss as the training process, then the loss $J(\alpha)$, predicted class probabilities $\mathbf{y}(\alpha)$ and the true labels $\mathbf{y}^*$ are like:

$$J_{val}(\alpha) = \frac{1}{N} \sum_{n=1}^{N} \text{cross-entropy}(\mathbf{y}_i(\alpha), \mathbf{y}_n^*)$$

where $N$ represents the number of validation samples. If the validation loss is of this case, in each training epoch $t$, we are able to do gradient search over continous loss function on evaluation set, over several candidate $\alpha$ and identify the one that minimizes the evaluation error. In other words, we compute candidate scale factors $\alpha_1 = 0.5\alpha_t$, $\alpha_2 = \alpha_t$, and $\alpha_3 = 2.0\alpha_t$ and select the optimal scale factor $\alpha_{t+1}$ such that:

$$\alpha_{t+1} = \arg \min_{\alpha \in \{\alpha_1, \alpha_2, \alpha_3\}} J(\alpha).$$

But in general case, the criterion may be discrete, such as 0/1 loss on a set of questions, such as multiple choices, numerical questions or translation questions.

Suppose our custom task is to learn the recent paper on Neurips conference and related code. The multiple choices are like as MMLU (Massive Multitask Language Understanding), e.g., *Question: Which paper first proposed the Transformer architecture? Answer Options: A. "Attention Is All You Need"; B. "Deep Residual Learning for Image Recognition".*

Numerical questions are like GSM8K (Grade School Math 8K), e.g. *Question: What is the parameter size of the last fully-connected layer in AlexNet? Answer: Assume the input size to the fully connected layer is* 4096 *and the output size is* 1000. *Solution Steps:...* $n_{in} = 4096$ *and* $n_{out} = 1000$. *Calculate the number of parameters:* $4096 \times 1000 + 1000 = 4096000 + 1000 = 4097000..$

The output tokens of a LLM need to exactly match the option A and the value 4097000, an exact match generate a loss of 0, and a loss of 1 otherwise. For other tasks, such as translation, the criterion could be based on metrics such as BLEU (Bilingual Evaluation Understudy) score, which measures the similarity between the generated translation and a reference translation.

The evaluation should consider both loss on customization tasks, and on standard general tasks which measure pretraining models. The ideal case is that the overall evaluation loss is quasi-convex w.r.t the scale factor in within these sampled $\alpha$, i.e.

$$J(\lambda \alpha_1 + (1 - \lambda)\alpha_2) \leq \max\{J(\alpha_1), J(\alpha_2)\} \tag{5}$$

for all $\alpha_1, \alpha_2 \in [\alpha_m, \alpha_M]$ and $\lambda \in [0, 1]$. In this case we have an optimal scale factor $\alpha^*$ that maximizes (or minimizes) the performance, so that as we move away from $\alpha^*$ in either direction, the performance degrades monotonically. If this is not the case, we have to find a balance point between performance on general tasks or for customization. Starting with $\alpha = 1.0$ which prioritize customization, the method iteratively decreases $\alpha$ toward zero. The goal is to identify a balance point. If decreasing $\alpha$ any further improves performance on generalized validation questions but degrades performance on customized tasks, the search terminates, and the optimal $\alpha$ is determined. This balance point allows the user to make an informed decision about the tradeoff between generalization and customization based on their specific requirements. This procedure is showed in Fig 2.

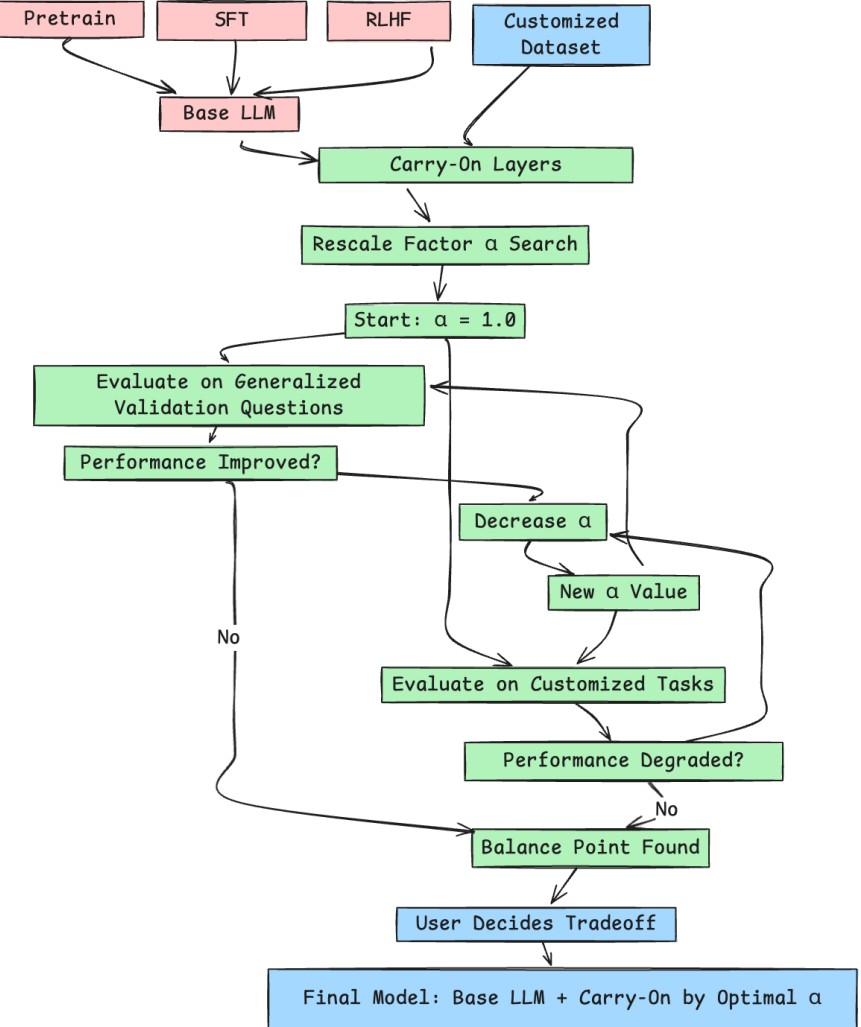

Figure 2: Optimal carry-on model search of customization pipeline.

## 5 Statistical Theory

We notice that this carry-on approach, regardless of the architectural and system compoenents, train the customized LLM on three datasets. First the base LLM $h_w$ was trained on massive scale pretraining corpus and supervised fine-tuning datasets, and possibly reinforcement learning pipeline afterwards. The the main component of carry-on models, $f_\theta$ and gate $g_\theta$ are trained on customized training data, and we further search optimization $\alpha$ on the validation set. We try to analyze the generalization performance of this kind of machine learning pipeline, which is fundamentally governed by the tradeoff between model complexity and the amount of training data. A key theoretical framework for understanding this tradeoff is the Vapnik-Chervonenkis (VC) dimension Vapnik & Chervonenkis (1971), which quantifies the capacity of a function class to fit arbitrary patterns in the data. Models with high VC dimension are prone to overfitting when training data is limited. Let $\mathcal{H}_W$ be the function class for the model parameters $W$ of the carry-on layers, $\mathcal{H}_\alpha$ be the function class for the scale factor $\alpha$. Since $\alpha$ is a scalar, its VC dimension is $d_{\mathrm{VC}}(\mathcal{H}_\alpha) = 1$. In contrast, $W$ is a high-dimensional matrix, and $d_{\mathrm{VC}}(\mathcal{H}_W)$ is typically much larger. When $W$ and $\alpha$ are jointly trained on the training set, their VC dimension $\mathcal{H}_{W,\alpha}$ satisfies:

$$d_{\mathrm{VC}}(\mathcal{H}_{W,\alpha}) \geq d_{\mathrm{VC}}(\mathcal{H}_W) + d_{\mathrm{VC}}(\mathcal{H}_\alpha) = d_{\mathrm{VC}}(\mathcal{H}_W) + 1.$$

By decoupling $W$ and $\alpha$, we effectively reduce the complexity of the function class. Specifically, $W$ is trained on the training set, and its generalization error is bounded by:

$$\mathcal{E}_{\text{gen}}(W) \leq \mathcal{E}_{\text{train}}(W) + \mathcal{O}\left(\sqrt{\frac{d_{\text{VC}}(\mathcal{H}_W) + \log(1/\delta)}{N_{\text{train}}}}\right).$$

Since $d_{\text{VC}}(\mathcal{H}_\alpha) = 1$ is negligible compared to the first, leading to a tighter overall bound. $\alpha$ is optimized on the validation set, and its generalization error is bounded by:

$$\mathcal{O}\left(\sqrt{\frac{d_{\text{VC}}(\mathcal{H}_\alpha) + \log(1/\delta)}{N_{\text{val}}}}\right).$$

When $\alpha$ is trained on the training set, it becomes part of the optimization process for $W$. This introduces optimization bias, as $\alpha$ is tuned to minimize the training loss, which may not generalize well to unseen data. By contrast, optimizing $\alpha$ on the validation set ensures it to minimize the validation loss, which is a better proxy for generalization.

The most general approach (training $W$ and $\alpha$ on the training set) introduces additional variance due to the joint optimization of $W$ and $\alpha$. This increases the risk of overfitting, as the model can "memorize" the training data by adjusting both $W$ and $\alpha$. The more complex approach mitigates this by isolating $\alpha$-optimization on the validation set, reducing variance and improving generalization.

## 6    Gradient Boosting Perspective

The framework adds additional transformer layers on top of the highest layer of an existing LLM and to search the optimal scale factor $\alpha$ for these new layers, can be viewed as an iterative process of gradient boosting Friedman (2001) where each new layer acts as a "weak learner" that refines the predictions of the base model (the pretrained LLM). At each iteration $t$ of gradient boosting, a new weak learner $h_t$ is trained to fit the negative gradient of the loss function with respect to the current model's predictions. The final model is a weighted sum of the base model and the weak learners:

$$F(x) = h(x) + \sum_{t=1}^{T} \alpha_t f_t(x),$$

where $h(x)$ is the base model, $f_t(x)$ are the weak learners, and $\alpha_t$ are the step sizes (scale factors).

In the proposed framework: The pretrained LLM serves as the base model $h(x)$. The additional transformer layers act as weak learners $f_t(x)$, refining the predictions of the base model. The scale factor $\alpha$ controls the contribution of the new layers, analogous to the step size in gradient boosting. Let $\mathcal{L}(y, F(x))$ be the loss function, where $y$ is the target output and $F(x)$ is the model's prediction. In gradient boosting, the negative gradient $g_t(x)$ at iteration $t$ is:

$$g_t(x) = -\frac{\partial \mathcal{L}(y, F_{t-1}(x))}{\partial F_{t-1}(x)}.$$

the step size $\alpha_t$ is often chosen to minimize the loss on the training data:

$$\alpha_t = \arg\min_\alpha \mathcal{L}(y, F_{t-1}(x) + \alpha g_t(x)).$$

Using the statistical theory of gradient boosting, we can derive generalization bounds for the proposed framework. Let $\mathcal{H}_t$ denote the function class for the $t$-th additional layer. The VC dimension of $\mathcal{H}_t$ is typically much smaller than that of the base model, as the new layers are shallow and operate on residual signals. Similar to Bartlett & Mendelson (2002), the generalization error of the final model $F_T(x)$ is bounded by:

$$\mathcal{O}\left(\sqrt{\frac{\sum_{t=1}^{T} d_{\text{VC}}(\mathcal{H}_t) + \log(1/\delta)}{N_{\text{train}}}}\right),$$

where $d_{\text{VC}}(\mathcal{H}_t)$ is the VC dimension of the $t$-th additional layer. Since $d_{\text{VC}}(\mathcal{H}_t)$ is small, the bound remains tight even as new layers are added.

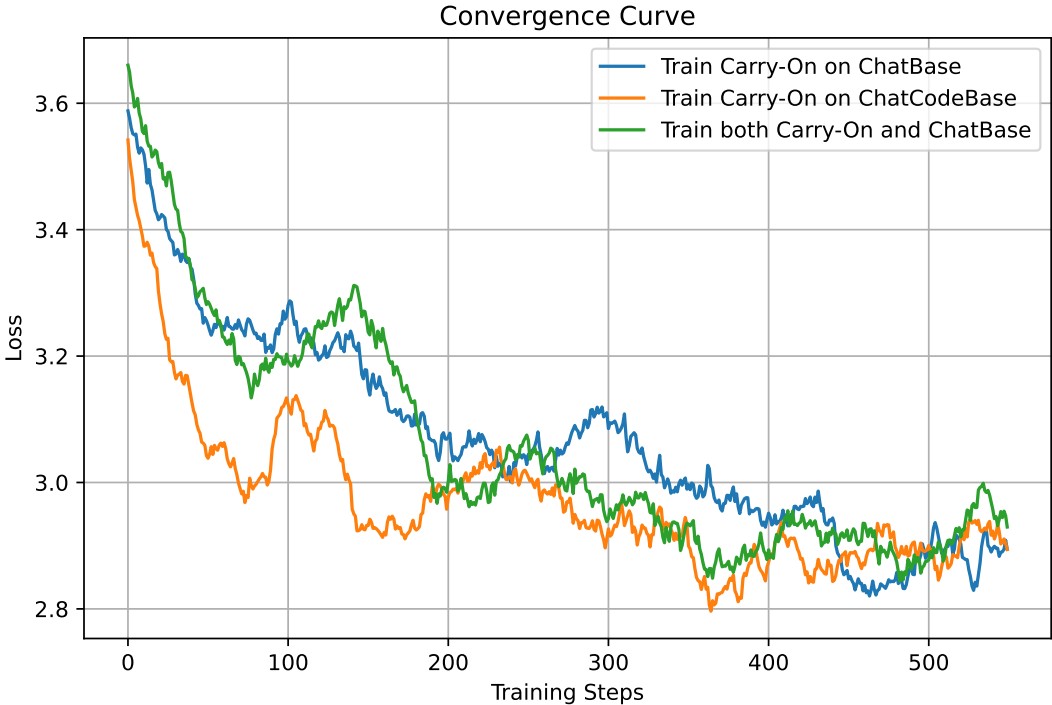

Figure 3: The convergence comparison between training on the carry-on wi/wo training base LLM.

## 7 Experiments

We study the loss function convergence and intelligence emergent capability when training the carry-on. The first study reflects metrics important to pretraining, and the second study aims at supervised fine-tuning or complex reasoning. We conduct experiments with several Qwen2.5 model variants Yang et al. (2024) as base models, e.g. Qwen2.5-Coder-Instruct, optimized for code generation; Qwen2.5-Instruct, a general purpose LLM. In addition we test Marco 7B which is continue-pretrained from Qwen for reasoning, and Moonlight is a small DeepSeek-V3 MoE Model Liu et al. (2024). We use MoE transformer and a tiny two layer neural network as carry-on.

In another pretraining experiment, we built a 5 layer MoE transformer of the DeepSeek-V3 architecture on top of quantized Qwen2.5 7B with AWQ quantization Lin et al. (2024). The DeepSeek-V3 architecture integrates Multi-Head Latent Attention (MLA), using both positional encoding (RoPE) and non-positional encoding (NoPE) for query-key computations. The MoE layer routes each token to 8 out of 32 experts, enabling the model to specialize in different aspects of the input data. This routing mechanism enhances the model's capacity without a proportional increase in computational cost. We perform a comparison of training entire base model plus carry-on, v.s. only training the carry-on; and a comparison between training a carry-on on one base model (Chat 7B) or two base models (Chat 7B and Coder 3B). The internal embedding dimension of the carry-on transformer is 1024, so a projection from 3584 dimension (7B) or from 2048 dimension (3B) is needed; and they are both needed if two base models are used, where we average two branch embedding projection by 1:1. We add a bottleneck linear layer of 128 dimensional embedding, and project it to vocabulary space, which saves parameters of a much larger projection. The configuration of the DeepSeek-V3 model is summarized in Table 1.

Table 1: DeepSeek-V3 Architecture Carry-On Layers

| Parameter | Value |
|---|---|
| Num Hidden Layers | 6 |
| Hidden Size | 1024 |
| Expert Hidden Size | 256 |
| KV LoRA Rank | 128 |
| Q LoRA Rank | 128 |
| Num Attention Heads | 32 |
| Num KV Heads | 8 |
| QK Dim (NoPE) | 32 |
| QK Dim (RoPE) | 16 |
| Value Head Dim | 32 |
| Num Routed Experts | 32 |
| Experts per Token | 8 |
| Num Shared Experts | None |
| Dense Layers Replaced | None |

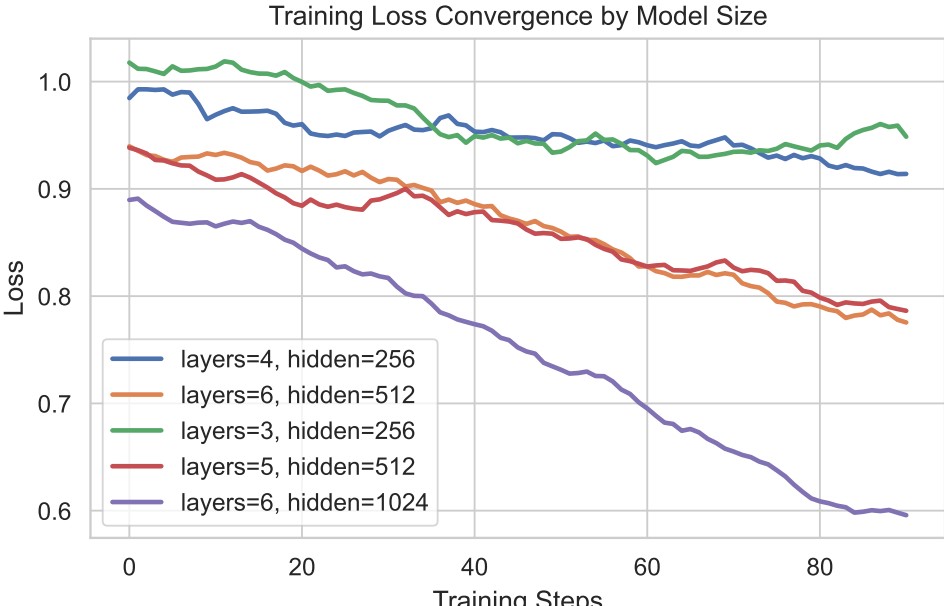

Figure 4: Training convergence with different carry-on layer size (quantize bits=4, shallow shortcut layer depth = 0).

## 7.1 Rebase DeepSeek-V3 on Qwen2.5

This experiments follows a pretraining style data collection. We train the model with 100,000 samples of Cosmopedia data, which was generated by Mixtral LLM to cover topics of textbooks, WikiHow and blogs; we add 100,000 pieces of OpenWebText samples; and we add 300,000 samples of Magpie extraction of Qwen 72B LLM.

In the first case, the model is cast to `torch.bfloat16` for training and for the second case AWQ quantization is enabled The input embeddings of the target model are replaced with those from the Qwen base model, and the model is trained using a cosine learning rate scheduler with warmup steps. The training process runs for 2 epochs. We set the learning rate for carry-on to be 1e-4, and make the learning rate of base model 1e-5 if we train it. On top of DeepSeek MoE, we implemented an easier version of experts router, which adds a dropout layer and the dropout probability is gradually reduced from 0.5 to 0.1 throughout the training. GPU memory

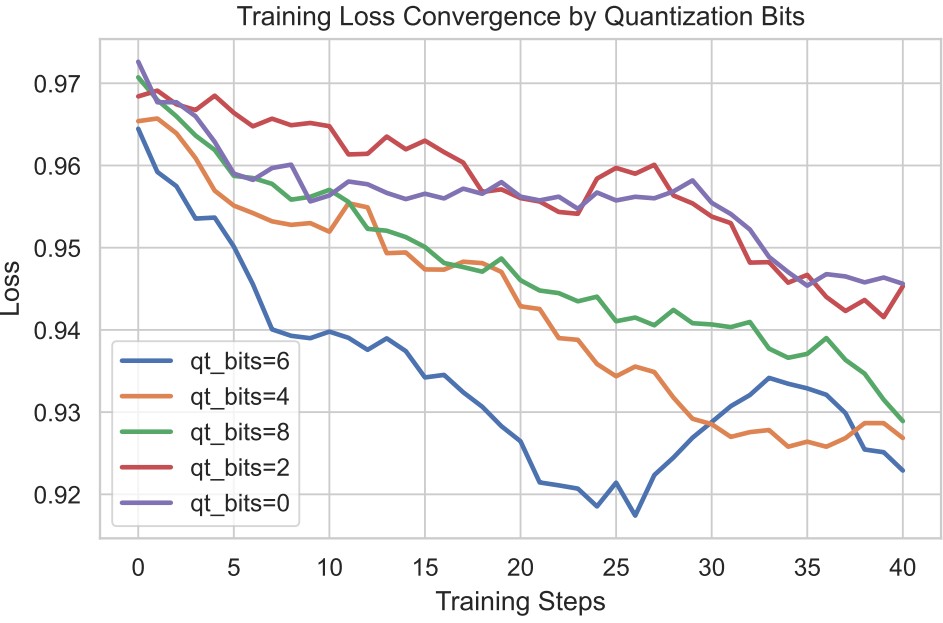

Figure 5: Training convergence with different quantization bits (qt_bits), and qt_bits = 0 means no quantization to the floating point embedding. (shortcut layer depth=0, hidden size = 256, carry-on layer = 3)

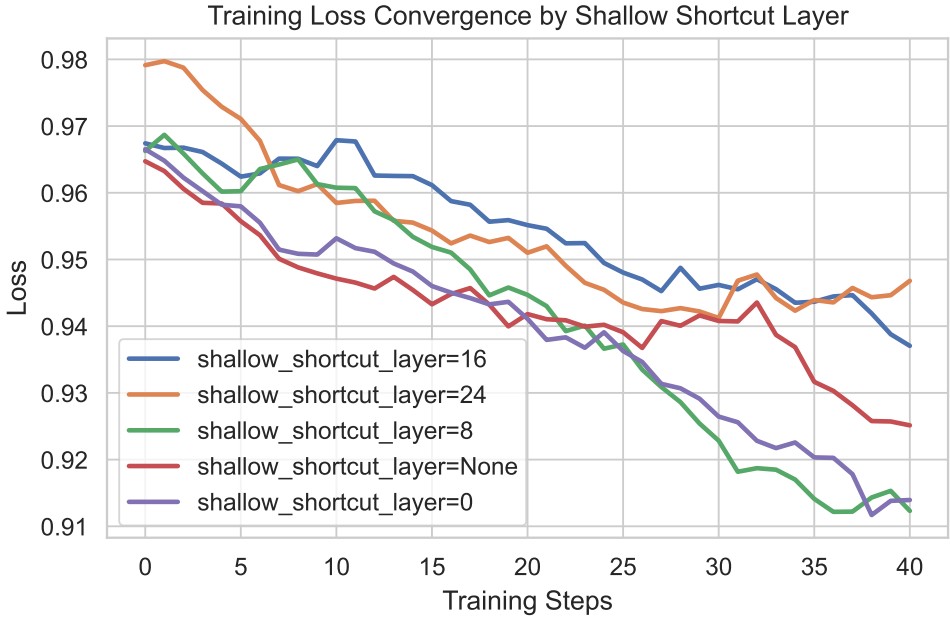

Figure 6: Training convergence with different shortcut shallow layers (quantized bits=4, hidden size = 256, carry-on layer = 3).

usage is logged periodically to monitor resource utilization. The MoE router implementation dynamically selects the top-$k$ experts based on their scores during training, ensuring efficient routing of tokens to the most relevant experts. We can see from convergence curve from Fig3 that a carry-on based on two Qwen models, chat and coder, convergence faster than itself based on only chat model; and training both carry-on and base model doesn't necessarily improve convergence speed, which indicates too many degrees of freedom to result in side effects.

We use 3000 piece of the the long-cot query-answer pairs and try to test whether bigger and more carry-on layers helps to memorize this certain dataset quickly and accurately enough, which is important to LLM customization to certain working fields with specialized terminology and narratives. By varying number of layers and the dimensions of each layer embedding, we plot the loss convergence rate in Fig.(4). We empirically found that there is a scaling-law for the carry-on transformers, when the base LLM fixed.

We compare different embedding quantization strategy, try to compress the floating point number to smaller bits, to save communications, and we plot the results in Fig.(5). We see that floating point representation of original embedding are not always needed, since some quantized embeddings helps to converge even faster, although more bits do benefit. We compared different strategy for take shallow layers and fuse into the highest layer embeddings, and plot the convergence in Fig.(6, where we see that introducing shallow layers accelerate training, but higher layers (e.g. 16, 24) don't contribute as good as shallow layers (e.g. 0, 8), proving that shallow layers provide orthogonal complementary information to the higher layers.

## 7.2 Tiny CarryOn for Math Reasoning

Table 2: Performance comparison across model size and specialties, on GSM8K accuracy (Acc) and validation cross-entropy loss.

| Model | Val Loss | Acc(Base) | Acc(CarryOn) |
|---|---|---|---|
| Chat3B | 0.88 | 26% | 46% |
| Chat7B | 0.89 | 56% | 60% |
| Coder3B | 1.01 | 1% | 13% |
| Coder7B | 0.83 | 15% | 36% |
| Moonlight3B | 1.01 | 8% | 11% |
| Marco7B | 0.77 | 63% | 70% |

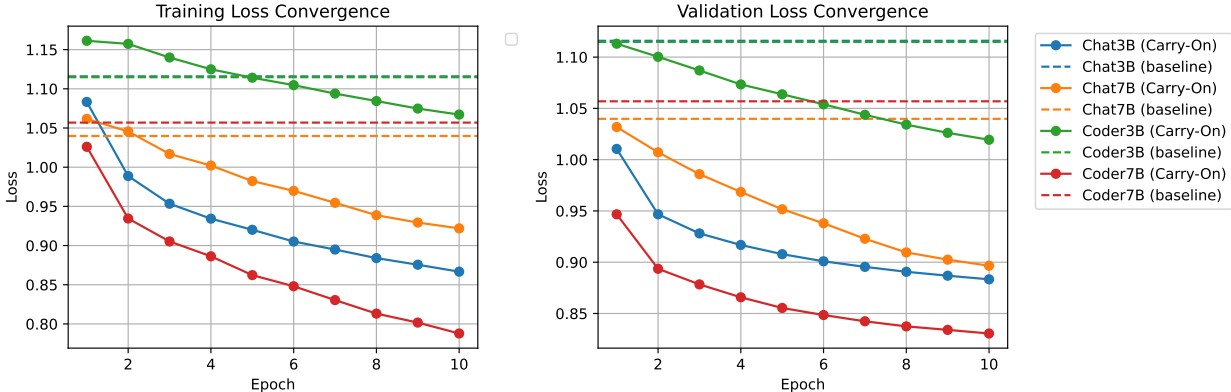

Figure 7: Loss convergence on Qwen LLMs to fit complex reasoning chain-of-thoughts like next-token-prediction.

We experimented with adding extremely small carry-on of two linear layers, the first layer projects to 64 dimensions and the second layer projects back to the original dimension. This carry-on reuses the base LLM decoding header, so it is less than 1MB in parameter size. We truncate each data sequence by maximum of 512 tokens, and start next-token prediction from the 30-th token, making the prior words as prompts. Models are trained with the AdamW optimizer ($\beta_1 = 0.9, \beta_2 = 0.95$). A key innovation in our approach is the dynamic optimization of the scale factor. The initial Value is 0.1. At each epoch, we evaluated multiple rescale factors ( 0.3, 0.5, 1.0, 2.0, 3.0) on the validation set. The scale factor yielding the lowest validation loss was selected for the next epoch. If the optimal rescale is 1.0, we use the square-roots of these factors to narrow down the search. During training, we evaluated our models using the cross-entropy loss, both measured on validation sets and training set to assess language modeling quality, showed in Fig7.

We evaluate the performance of two versions of our model: the original model and an enhanced version that incorporates a residual predictor. The evaluation process involves generating answers to the math problems in the GSM8K dataset and comparing the model's predictions to the ground truth answers. We select the first 100 samples from the test set for evaluation. To ensure consistent answer formatting, we use a predefined prompt template that instructs the model to provide the final numerical answer after the **####** delimiter. The template is as follows: *Math Question: {question} Let's analyze and solve the question, but don't write program code, and write the final number results after ####. Examples: after calculattion, the square footage is #### 1000 square feets.* This evaluation is somehow harder than standard tests, as the model is not used in chat mode by this prompt, which doesn't have system prompts with keywords like *role: user, assistant, system* and *content*, and the example sentence in the prompt could mislead the answer to follow to be 1000. For each math problem, we generate answers using both the original model and the enhanced model. The maximum number of tokens to generate is set to 800. Sampling is disabled to ensure deterministic outputs.

The model's response is parsed to extract the numerical answer following the **####** delimiter. We use a regular expression to search for all occurrences of numerical values preceded by the **####** delimiter. The regular expression `r'####\s*(-?\d+(\.\d+)?|\d+/\d+)'` is used to match integers, floating-point numbers, and fractions. It filters out any empty matches and selects the last valid numerical value in the text, as this is assumed to be the final answer. The extracted value is converted to either an integer or a float, depending on its format.

We calculate the accuracy of both the original and enhanced models by comparing the predicted answers to the ground truth answers. The accuracy is defined as the ratio of correctly answered questions to the total number of questions evaluated. In Table 2 we see that this 1MB level carry-on is able to help a small LLM to emerge reasoning capability. In comparison, chat models under small footprints is easier to train than coder models.

## 8 Conclusion

Our work demonstrates that an easier and scalable training framework of stacking deeper transformer layers can utilize inference GPU servers for customization training, mixing existing state-of-the-art models from different expertise, and there exists a better way to control customization scale against overfitting. Without relying on high-end GPU to train, it is affordable to personal use or small tasks. We look forward to more techniques for similar objectives to be industrialized to benefit more users to have a larger degree-of-freedom to tune their own version of AI tools.

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
