# OpenReview forum: "GPT Carry-On:  Language Model Customization Made Scalable by Growing-In-Depth"
_TMLR — Rejected by TMLR_

### Review · Reviewer_gtR9 · 2025-06-07

**Summary Of Contributions:**

This paper presents "GPT Carry-On," a novel framework for customizing large language models (LLMs) that addresses the computational challenges of personalizing models for individual users or specific tasks. The key contributions are:

1. A distributed training architecture that separates inference and training computations across different GPU nodes, allowing the use of existing inference infrastructure for customization tasks. The base model performs forward passes on inference nodes (potentially with quantization), while lightweight "carry-on" transformer layers are trained on separate training nodes.
2. A modular approach to LLM customization that trains additional transformer layers on top of frozen pretrained models, using a learnable scale factor α and gating mechanism to balance between base model knowledge and task-specific adaptation.

**Audience:**

Yes

**Broader Impact Concerns:**

N/A.

**Claims And Evidence:**

Yes

**Requested Changes:**

1. Include at least one experiment comparing GPT Carry-On with LoRA or QLoRA on the same task (e.g., GSM8K) using comparable parameter budgets. This comparison should report both accuracy metrics and training time/memory usage. Even a simple comparison on your existing experimental setup would help readers understand the relative merits of your approach.

2.  Increase the evaluation set from 100 to at least 500 GSM8K test samples to provide more statistically reliable results. If possible, also report confidence intervals or standard deviations across multiple runs. Additionally, consider testing on one more larger dataset o demonstrate generalization beyond GSM8K.

3. Add a brief analysis or empirical measurement of the network communication overhead between inference and training nodes. This could be as simple as reporting the data transfer size per training step and estimated bandwidth requirements for your experimental setup. Include a discussion of how this scales with batch size and model size to help readers assess practical deployment considerations.

**Strengths And Weaknesses:**

### Strengths

1. The framework cleverly leverages existing inference infrastructure, addressing real-world deployment constraints where inference nodes are optimized for low-latency serving rather than training.
2. The ability to mix multiple base models and control the contribution of carry-on layers through learnable scale factors provides flexibility in balancing generalization and customization.
3. The connection to gradient boosting and statistical learning theory provides a solid foundation for understanding why the approach works.

### Weaknesses
1. While the paper mentions differences from LoRA and other parameter-efficient fine-tuning approaches, it lacks direct performance comparisons with these methods under similar computational budgets. This is a critical omission as readers cannot assess whether GPT Carry-On offers advantages over established PEFT techniques.
2. The math reasoning experiments use only 1,000 training samples and evaluate on 100 test samples. Such limited scale makes it difficult to conclusively demonstrate the emergence of reasoning capabilities or generalization to complex tasks, like MMLU, Hellaswag, etc.
3. The paper doesn't thoroughly analyze the network communication costs between inference and training nodes, which could be significant in distributed settings.

---

### Review · Reviewer_2dgt · 2025-06-12

**Summary Of Contributions:**

The paper targets the question of reducing costs of LLM customization for users and tasks and proposes a brand new and novel carry-on-training framework. The framework echoes the problem by introducing additional architectures atop existing base models and separating training and inference in different nodes. The paper serves as basic concept proof for the framework and the research question, though tons of continuing industrial practices are needed to validate the effectiveness of the paradigm.

**Audience:**

Yes

**Claims And Evidence:**

Yes

**Requested Changes:**

1. Rewrite the paper to make it more readable and spotlight its logic, especially in the Experiment section.
2. Add necessary experiments to support claims in the former part, including more datasets, different carry-on architectures, the computation usage and time costs, comparison with LoRA.
3. All citations in the paper use wrong latex format: "The rapid development of large language models (LLMs) like GPT-4 Achiam et al. (2023) and DeepSeekR1Guo et al. (2025)..."

**Strengths And Weaknesses:**

Strengths
1. The question is interesting and the framework is novel. Statistical Theory and Gradient Boosting Perspective are conducted to support the framework.
2. The framework is well formalized and clarified in the first 6 sections (except for the Experiment section).

Weaknesses
1. The paper is hard to read. Logic transitions and summarization are missing in paragraphs, sections and throughout the paper. It is especially chaotic in the Experiment section. It is hard for the reader to grasp the setting and what the comparison is for.
2. The experiment can not support the general framework. The datasets used are insufficient since the paper aims at model customization on various tasks and users, failing to support the effectiveness of the framework. And LoRA may be conducted as baseline, since the framework updates small gradients as well.

Overall, I think the paper needs substantial modifications for submission.

---

### Review · Reviewer_Dhxi · 2025-07-05

**Summary Of Contributions:**

This paper proposes a novel framework for customizing LLMs by adding lightweight transformer layers on top of pre-trained models. The authors aim to address the challenge of personalizing LLMs for individual users or specific tasks without requiring extensive computational resources. The proposed method involves training additional transformer blocks on the final-layer embeddings of pre-trained LLMs, allowing for the creation of customized models with minimal computational overhead. The paper demonstrates the effectiveness of this approach through experiments on various LLM variants and tasks, showing promising results in terms of loss convergence and task-specific performance.

**Audience:**

Yes

**Claims And Evidence:**

Yes

**Requested Changes:**

Please refer to the weakness part.

1. Some more experiments are expected.

2. The writing needs to be improved, and some doubts with LoRA need to be clarified.

**Strengths And Weaknesses:**

Strengths

1. The paper presents a scalable method for customizing LLMs by adding lightweight transformer layers, which can be trained with minimal computational resources. This approach allows for the creation of task-specific models without requiring high-end GPUs, making it accessible for a broader range of users.

2. By leveraging inference nodes for forward passes and training nodes for lightweight carry-on layers, the proposed framework efficiently utilizes computational resources. This separation of tasks reduces the memory and computational burden on training nodes, enabling faster and more efficient training.

3. The authors introduce a very simple and intuitive optimization process for the scale factor controlling the contribution of the carry-on layers. The simplicity of the method is a very good point of the paper.


Weakness


1. The paper primarily evaluates the proposed method on specific tasks such as math reasoning and pretraining tasks. However, it lacks a comprehensive evaluation on a diverse set of tasks that cover different domains and complexities. For example, the paper does not include evaluations on natural language inference, sentiment analysis, or machine translation tasks. A broader evaluation would provide a more complete picture of the method's effectiveness across various applications.

2. The writing may also be improved. Several Figures are too big, making the paper somewhat redundant. Also, the algorithmic plot is just a Snapchat-like format, not the LaTeX format.

3. How does the author choose the specific architecture in the carry-on? What's the benefit compared to LoRA if the architecture also needs to be carefully chosen in the carry-on algorithm?

---

### Decision · Action_Editor_XKuw · 2025-09-25

**Recommendation:** Reject

**Audience:**

Yes

**Audience Explanation:**

The proposed methodology is efficient, simple, and effective. I think the method should be beneficial to the LLM community. But unfortunately, due to the flaws in the paper, I have to reject it.

**Claims And Evidence:**

No

**Claims Explanation:**

According to the reviewers, the authors did not provide sufficient experiments to support the claims in the paper. Firstly, comparison with more baselines should be conducted. Secondly, more experiments in different domain settings should be added.